# Focus on Formononetin: Anticancer Potential and Molecular Targets

**DOI:** 10.3390/cancers11050611

**Published:** 2019-05-01

**Authors:** Samantha Kah Ling Ong, Muthu K. Shanmugam, Lu Fan, Sarah E. Fraser, Frank Arfuso, Kwang Seok Ahn, Gautam Sethi, Anupam Bishayee

**Affiliations:** 1Department of Pharmacology, Yong Loo Lin School of Medicine, National University of Singapore, Singapore 117600, Singapore; samantha.ong@u.nus.edu (S.K.L.O.); phcsmk@nus.edu.sg (M.K.S.); phcfanl@nus.edu.sg (L.F.); 2Lake Erie College of Osteopathic Medicine, Bradenton, FL 34211, USA; SFraser78637@med.lecom.edu; 3Stem Cell and Cancer Biology Laboratory, School of Pharmacy and Biomedical Sciences, Curtin Health Innovation Research Institute, Curtin University, Perth, WA 6102, Australia; frank.arfuso@curtin.edu.au; 4Department of Science in Korean Medicine, Kyung Hee University, 24 Kyungheedae-ro, Dongdaemun-gu, Seoul 02447, Korea

**Keywords:** formononetin, cancer, preclinical models, cell signaling, angiogenesis

## Abstract

Formononetin, an isoflavone, is extracted from various medicinal plants and herbs, including the red clover (*Trifolium pratense)* and Chinese medicinal plant *Astragalus membranaceus.* Formononetin’s antioxidant and neuroprotective effects underscore its therapeutic use against Alzheimer’s disease. Formononetin has been under intense investigation for the past decade as strong evidence on promoting apoptosis and against proliferation suggests for its use as an anticancer agent against diverse cancers. These anticancer properties are observed in multiple cancer cell models, including breast, colorectal, and prostate cancer. Formononetin also attenuates metastasis and tumor growth in various in vivo studies. The beneficial effects exuded by formononetin can be attributed to its antiproliferative and cell cycle arrest inducing properties. Formononetin regulates various transcription factors and growth-factor-mediated oncogenic pathways, consequently alleviating the possible causes of chronic inflammation that are linked to cancer survival of neoplastic cells and their resistance against chemotherapy. As such, this review summarizes and critically analyzes current evidence on the potential of formononetin for therapy of various malignancies with special emphasis on molecular targets.

## 1. Introduction 

Cancer refers to the disease that develops when abnormal cells proliferate uncontrollably, followed by invasion into the surrounding tissues and eventually spreads to the organs or other parts of the body via the circulatory and lymphatic systems [1]. The fundamental process that leads to the development of cancer is the process of continuous, unregulated proliferation of cancer cells, resulting in tumor development [2]. Cancer has claimed the lives of 8.8 million individuals in 2015, making it the second leading cause of death worldwide after cardiovascular diseases [3,4]. In the past few decades, numerous treatment methods have been developed against cancer after acquiring a deeper understanding of multiple underlying signals and mechanisms that contribute to the survival and progression of neoplastic cells. These treatment modalities include both adjuvant and neoadjuvant chemotherapy, targeted therapy, immunotherapy, surgery, and radiotherapy. The incidence of cancer still remains high, with increasing mortality due to the disease progression despite significant progress in treatment regimens [3,4]. This phenomenon can be largely attributed to the limited effects exhibited by existing cancer therapies and the high cost of treatment, coupled with significant adverse effects [5]. Furthermore, conventional cytotoxic agents usually embody life-threatening toxicities [6]. There have been nearly 500 cases of withdrawal of medicinal products from the market as a result of adverse drug reactions over the past six decades, with the most common reason being hepatotoxicity [7]. In addition, certain cancers, such as breast cancer, can still resurface after a dormant period of 15 years following successful treatment [8], suggesting that there is a need to develop new and safe treatment options that can be proven to be more efficacious. 

Natural products and dietary agents have gained enormous popularity over the years to be used in the clinical setting. They can be obtained from plants, microorganisms, or animals and have been used medicinally for decades, usually in complementary and alternative medicine [9]. An example would be Veregen, a topical ointment containing green tea extract sinecatechins, which is used to treat external genital and perianal warts, and the herb *Artemisia annua*, from which pharmaceutical scientist Tu Youyou managed to isolate artemisinin that is now used as a potent antimalarial drug [10]. These studies indicate the promising use of natural products for future discovery and development of cancer preventive and anticancer drugs. In the past three decades, nearly 80% of the drugs approved by the United States Food and Drug Administration for cancer therapy contain natural products or imitate their functions [11,12]. Natural products have since then made their appearance and represent a large portion of today’s pharmaceutical agents used in cancer therapy, including taxol, vinblastine and camptothecin [13]. Over 60% of the current anticancer drugs are derived from natural sources [13], and the impetus to discover more natural products for chemotherapy and chemoprevention has become evident in the past decade as more agents are now in clinical trials.

Antitumorigenic activity can be found in compounds with different structural groups, including isoprenoids (including terpenoids and carotenoids), isoflavones, etoposide, and teniposide [5,11,14,15,16,17,18,19,20,21,22,23,24,25,26,27,28,29,30,31,32,33,34,35]. They function through various mechanisms, including the induction of apoptosis via DNA cleavage by inhibiting the activity of topoisomerase I or II, mitochondrial permeabilization, inhibiting crucial enzymes in signal transduction (i.e., proteases), cellular metabolism, or by inhibiting tumor-induced angiogenesis [9,36]. Certain isoflavones, such as isoliquiritigenin, have been shown to possess antitumorigenic properties, including pro-apoptotic effect on human cancer cells [37]. In recent years, there has been increasing evidence in preclinical and clinical studies that suggest that a dysregulated inflammatory response plays a crucial role in cancer development and may even be a key driver of cancer [38,39]. 

Chronic inflammation is also one of the hallmarks of cancer, and can drive the development of cancer through increasing the production of pro-inflammatory mediators involved in various signaling mechanisms, including cytokines, chemokines, reactive oxygen species/intermediates, increased expression of oncogenes, cyclooxygenase-2, 5-lipoxygenase, matrix metalloproteinases (MMPs), and pro-inflammatory transcription factors, such as nuclear factor‑κB (NF-κB), signal transducer and activator of transcription 3 (STAT3), activator protein 1 (AP-1) and hypoxia-inducible factor 1α (HIF-1α) [38,39,40,41,42,43,44,45,46,47,48,49,50,51,52,53,54,55,56,57,58,59,60,61,62,63,64,65,66]. These factors mediate the basis of cancer progression, such as the proliferation of tumor cells, metastasis, survival, invasion, angiogenesis, chemoresistance and radio-resistance [38,47,61,64,67,68,69,70,71]. 

Targeting selected transcription factors and pathways increases the sensitivity of cancer cells to chemotherapeutics and radiation, resulting in apoptosis [60]. Furthermore, the use of natural products, such as farnesol, curcumin and celastrol, have been found to significantly enhance the anticancer effects of chemotherapeutics, such as bortezomib and thalidomide, in multiple myeloma [57,72]. This suggests that natural products can be used as adjunct cancer therapy. Despite the steady improvement of current anticancer therapeutics, developing novel drugs remains a priority of cancer treatment due to an overwhelming increase in resistance to chemotherapeutic drugs [9,73,74,75,76]. With greater knowledge and constant advances in technology, there is a great prospect for safer and more efficacious treatment options for cancer. 

The red clover, *Trifolium pratense* (family: Fabaceae)*,* is a legume known for its numerous health benefits, and can hold a crucial role in the prevention and management of certain health conditions, including type 2 diabetes, hyperlipidemia and hypertension [77]. It is a perennial herb that is commonly found in Asia, Europe, and North America, and has been traditionally used to treat skin and respiratory conditions, such as eczema, psoriasis, asthma and pertussis [78]. The isoflavones present in red clover have estrogen-like activities and have been subjected to an intense research over the past two decades due to their potential cancer-preventive, cardio-protective and anti-osteoporosis effects [78]. 

The extract from the red clover plant contains genistein, daidzein, formononetin (biochanin B), and biochanin A. Formononetin [7-hydroxy-3-(4-methoxyphenyl)-4H-1-benzopyran-4-one], (Figure 1), one of the main bioactive components extracted from the red clover, has been found to be the principal compound that contributes the therapeutic effects observed in the extract. It is exclusively produced by the Fabaceae family, and can be extracted from the roots of *Astragalus membranaceus, T. pratense* and *Glycyrrhiza glabra* [79]. Since it is structurally similar to 17-estrogen, formononetin’s bioactivity mimics the effect of estrogen and this compound is considered to be a phytoestrogen [80]. 

Formononetin has shown beneficial effects in clinical trials for menopausal relief [81,82], reduction in bone loss [83], and improved arterial compliance [84]. Formononetin has been used clinically in China, in traditional medicine, as one of the fundamental herbs for treating carcinomas due to its protective effects against certain malignant tumors [85]. Formononetin has become the subject of intense research over the past decade due to its estrogenic effects and antitumorigenic properties. This review will summarize and critically analyze current evidence on the potential of formononetin for anticancer therapy with special emphasis on molecular targets.

## 2. Toxicity and Pharmacokinetics of Formononetin 

A water-soluble derivative of formononetin, formononetin-3′-sulphonate (Sul-F, C_16_H_12_O_7_SNa), has been shown to provide significant neuro- and cardio-protective effects both in vitro and in vivo [86,87]. Pro-estrogenic isoflavones, such as formononetin, can potentially be converted to more potent phytoestrogens in the human body. Incubation of formononetin in human liver microsomes caused demethylation, resulting in the production of formononetin derivatives and metabolites, including daidzein [88]. Due to the fact that formononetin is a naturally occurring isoflavone and phytoestrogen, it is associated with estrogen receptor (ER) binding. Phytoestrogens, especially isoflavones, can be classified as endocrine disruptors and are known to modify or interfere with the endocrine function [80]. 

Formononetin and its metabolites can significantly enhance pro-inflammatory cytokines and induce an allergic immune response. Interleukin-4 (IL-4) is closely associated with the CD4^+^ T helper cells and EL4 T lymphoma cells (*Mus musculus*). An increase in IL-4 production is observed upon administration of formononetin, daidzein, and equol due to an elevation in the activation of activator protein 1 (AP-1) via the phosphoinositide 3-kinase (PI3K)/protein kinase C (PKC)/p38 mitogen activated protein kinase (MAPK) signaling pathway. This suggests that formononetin and its metabolites may potentially cause allergic responses through amplifying the production of IL-4 in T-cells [89]. Hence, it may be a reasonable approach to restrict or limit the usage of formononetin in order to prevent allergic responses. 

## 3. In Vitro Anticancer Pharmacological Properties of Formononetin 

### 3.1. Antiproliferative Effects 

The anticancer potential of formononetin has been explored in numerous in vitro models as shown in Table 1. First and foremost, formononetin showed potential in inhibiting tumor growth and proliferation. Uncontrolled proliferation is one of the primary hallmarks of cancer and represents one of the most prominent factors associated with malignancy. This antiproliferative effect of formononetin has been observed in multiple cancer models and is one of the main anticancer properties of formononetin. As compared to other isoflavones, formononetin has been proven to possess the greatest antiproliferative activity [90]. The antiproliferative property of formononetin has been observed in ER-positive breast cancer cells, such as MCF-7 and T-47D [91,92], and displayed minimal effect against ER-negative breast cancer cells, namely MDA-MB-231 and MDA-MB-435 [93]. It also demonstrated antiproliferative effects against prostate cancer (PC-3, DU-145 and LNCaP) [94,95], non-small cell lung cancer (A549 and NCI-H23) [96], cervical cancer (HeLa) [97], bladder cancer (T24) [98], osteosarcoma (U2OS), ovarian cancer (ES2 and OV90) [99], glioma (C6), and colorectal cancer cells (HCT-116, SW1116 and RKO) [37,85].

For most cell lines, formononetin has been found to possess a concentration- and time-dependent effect against tumor proliferation [85,94,96,98]. Antiproliferative effects of formononetin were also demonstrated in cytokine-induced cancer models, such as multiple myeloma, where formononetin attenuated the expression of inflammatory cytokines, including tumor necrosis factor-α (TNF-α), transforming growth factor-β1 (TGF-β1), interleukin-6 (IL-6), and interleukin-8 (IL-8) by downregulating hypoxia-inducible factor 1α (HIF-1α) [39,100] and nuclear factor-κB (NF-κB) [101]. Most studies conducted thus far indicate that formononetin has been able to prevent the proliferation of tumor cells without causing serious adverse effects as compared to other chemotherapeutic drugs.

### 3.2. Proapoptotic Effects 

For progression of abnormal cells into cancerous ones, it is crucial that the apoptosis pathway is hijacked to allow promotion of growth and development of the damaged or abnormal cells. Cancer cells have the ability to evade the apoptotic checkpoints, allowing them to proliferate uncontrollably [102]. As a result, the proapoptotic property of formononetin elevates its potential to be used as an anticancer agent in cancer therapy. However, the underlying mechanisms of formononetin that promote cell apoptosis differ among different cell lines as well as cancer models. Apoptosis observed in cells can be classified into two different stages—early and late apoptosis—which can be differentiated through the presence of propidium iodide [102]. Formononetin largely elevated the proportion of early apoptotic cells in DU-145 prostate and U2SO osteosarcoma cell lines, and was found to be dose-dependent for prostate PC-3 cells (25–100 μM) [103,104,105]. 

The influence of formononetin on apoptosis of different cancer cell lines may involve the upregulation of specific transcription factors. The apoptotic mechanism in prostate DU-145 cells is activated by upregulating dexamethasone-induced retrovirus associated DNA sequences (Ras)-related protein 1 (maximum for 48 h before decreasing rapidly after) and Bcl-2-associated protein (Bax), and simultaneously reducing B-cell lymphoma 2 (Bcl-2) levels, thereby causing the DU-145 cells to display morphological changes indicative of the early apoptotic stage, and trigger apoptosis via the mitochondrial apoptotic pathway [106]. On the contrary, high concentrations of formononetin (>12.5 μM) have been found to effectively inhibit proliferation and trigger apoptosis of PC-3 prostate cancer cells by inhibiting the insulin-like growth factor 1 (IGF-1) receptor androgen-independent pathway [104]. T24 human bladder cancer cells displayed morphological changes of apoptosis when treated with formononetin, and there was a significant reduction in the expression of miR-21 and phosphorylated protein kinase B (AKT). In addition, phosphatase and tensin homolog (PTEN), a notable tumor suppressor gene, was upregulated in T24 cells after formononetin treatment, which suppressed uncontrolled tumor proliferation [98]. Furthermore, a study by Zhang and colleagues [107] suggested that formononetin did not elicit toxic effects on non-cancerous cell lines, indicating that it may be a safe choice to halt cancerous cell growth. 

One of the primary factors associated with cell apoptosis is the Bax and Bcl-2 protein levels, usually coupled with the AKT and extracellular signal regulated kinase (ERK) pathway. Bax protein is crucial for apoptosis in normal cells to prevent excessive proliferation and possible tumor formation. However, Bcl-2 functions in the opposite manner: enhancing cell survival by suppressing apoptosis [113]. Formononetin treatment reduced Bcl-2 protein levels and upregulated the pro-apoptotic proteins, such as Bax and caspase-3, thereby increasing the Bax/Bcl-2 ratio and inducing apoptosis. This effect was observed in several cancers, including prostate cancer (PC-3) [107], osteosarcoma (U2OS) [105], non-small cell lung cancer (A549 and NCI-H23) [96], and colon carcinoma (HCT-116) [37], and was sometimes coupled with caspase and phosphorylation activity. In PC-3 cells, increased phosphorylation of p38 and blocked AKT phosphorylation accompanied the growth in the Bax/Bcl-2 ratio to induce apoptosis [107]. An elevated activation and cleavage of caspase-3 was observed in A549 and NCI-H23 non-small cell lung cancer cell lines [96], and antiapoptotic proteins Bcl-2 and Bcl-xL were downregulated in HCT-116 colon cancer cells, along with caspase activation [37]. Furthermore, the upstream regulator and a novel pro-apoptotic protein, non-steroidal anti-inflammatory drug (NSAID)-activated gene (NAG-1) was found to be overexpressed in formononetin-treated HCT-116 cells, potentially promoting the apoptotic effects of formononetin; however, it failed to induce phase-specific cell cycle arrest [37]. In a recent study formononetin was found to inhibit the growth of osteogenic sarcoma U2OS cells and induce apoptosis by modulating the intracellular miR-375/ERα-PI3K/AKT signaling pathway [114].

Studies conducted thus far indicate that formononetin supports the apoptotic process of cancer cells through intrinsic pathways, but the mechanism and pathways may differ between different cancers. The most evident pathways suppressed by formononetin to induce apoptosis are the PI3K/AKT and the extracellular signal regulated kinase (ERK) pathways, which are the main regulatory pathways found dysregulated in many cancers. These pathways regulate the cell cycle and transmit signals from the surface of the cell to the nuclear DNA [115,116]. Some examples include cervical cancer (HeLa), osteosarcoma (U2OS) [90,105], and colorectal cancer (RKO) [101]. Other studies showed that formononetin successfully promoted apoptosis in prostate cancer (DU-145 and PC-3) [104,106,107], breast cancer (MCF-7 and MDA-MB-231) [92,93,108], non-small cell lung cancer (A549 and NCI-H23) [96], colon cancer (HCT116 and SW1116) [37,85], and nasopharyngeal carcinoma (CNE1 and CNE2) [112] via modulation of aforementioned pathways. Formononetin inactivated the MAPK signaling pathway, causing apoptosis in MCF-7 breast cancer cells and increased expression levels of Bax in LNCaP and PC-3 prostate cancer cells, leading to apoptosis [94]. Another commonly dysregulated pathway in cancer is the STAT3 signaling pathway. Emerging evidence suggests that abnormal STAT3 signaling drives the initiation and progression of human cancers through the inhibition of apoptosis and driving multiple pro-oncogenic functions [57,66,72,117,118,119,120]. Formononetin has been found to suppress fibroblast growth factor 2 (FGF2)-induced STAT3 activation, successfully suppressing multiple myeloma, leukemia, lymphoma and solid tumors that display constitutive STAT3 activation [121]. Furthermore, formononetin has been found to elicit greater antiproliferative and proapoptotic activity as compared to other isoflavones, such as calycosin [90]. 

### 3.3. Induction of Cell Cycle Arrest

Treatment of various cancer cell lines using formononetin has been shown to attenuate cancer cell proliferation via cell cycle arrest. Formononetin has been demonstrated to be highly effective in suppressing the oncogenic PI3K/AKT pathway [122,123,124,125,126,127,128,129,130,131,132] and inducing cell cycle arrest in numerous cell lines, including breast cancer (MCF-7), colorectal cancer (HCT-116 and SW1116) [85], non-small cell lung cancer [96], and prostate cancer (PC-3 and DU-145) [95]. Formononetin can induce cell cycle arrest at the G0/G1 phase by modifying the level of expression of cell cycle arrest-associated proteins and growth factors, as observed in DU-145 and PC-3 prostate cancer cells [95], A549 and NCI-H23 lung adenocarcinoma cells [96], ES2 and OV90 ovarian cancer cells [99], and MCF-7 breast cancer cells [91]. In non-small cell lung cancer, proliferation of A549 and NCI-H23 cells was significantly inhibited with treatment with formononetin. In human non-small cell lung cancer cells, formononetin induced cell cycle arrest in the G1-phase and promoted apoptosis via increasing p21 expression and reducing cyclin A and D1 expression in a time- and concentration-dependent manner [96].

Similarly, formononetin induced G1 arrest in PC-3 and DU-145 prostate cancer cells by downregulating cyclin D1, protein kinase B (AKT), and cyclin dependent kinase 4 (CDK4) in a concentration-dependent manner, and this phenomenon was observed to be more significant in PC-3 cells than DU-145 cells [95]. Inactivation of AKT facilitated the phosphorylation of glycogen synthase kinase-3β (GSK-3β), which, coupled with the reduction of cyclin D1, can deter progression into the G1/S phase [133]. Likewise, formononetin induced G0/G1 cell cycle arrest in MCF-7 cells by inactivating the IGF-1/IGFR-PI3K/AKT pathway [91]. On the contrary, cell cycle arrest in SW1116 and HCT-116 colon carcinoma cells at the G0/G1 checkpoint was induced through the suppression of cyclin D1 in a concentration-dependent manner, without changes in the expression levels of cyclin B1, which propagated the number of cells undergoing G1 phase cell cycle arrest to 79.7% upon treatment with 100 µM formononetin [85]. Collectively, these results suggest that the administration of formononetin modulated the expression levels of crucial cell cycle regulators through different pathways, consequently leading to cell cycle arrest at the G1 phase and apoptosis in various cancer cell lines. 

### 3.4. Antioxidant Effects

Formononetin has been investigated for its antioxidant properties and is known for being a potent and effective natural antioxidant capable of protecting cells from most free radicals. This is a crucial property of formononetin that may support its use as an anticancer agent because free radicals can lead to cancer development [134]. Formononetin is also the most effective isoflavone in inhibiting lipid peroxidation as it possesses the strongest antioxidant activity [135].

### 3.5. Angiogenesis-Modulating Effects 

Tumor angiogenesis is an essential pathway for the development and progression of malignant tumors [136], and studies have shown that the PI3K pathway may hold a vital role in this process [137]. It is a crucial process that leads to the growth and spread of cancer. The effect of formononetin on angiogenesis differs for different cancers. In one study, formononetin downregulated the expression of key pro-angiogenic factors, such as vascular endothelial growth factor (VEGF) and matrix metalloproteinases (MMPs), in LoVo human colorectal adenocarcinoma cells and reduced xenografted tumor size and the number of proliferating cells in the tumor tissues with decreased serum VEGF level [109]. MMP-2 and MMP-9 are known to be directly associated with tumor angiogenesis [138]. 

The most well-known pathway involved in antiangiogenic therapies currently under evaluation in clinical trials targets the VEGF pathway. However, there is a chance of the tumor acquiring resistance to the VEGF-targeted therapy by shifting to other angiogenesis mechanisms [121], therefore rendering the antiangiogenic treatment ineffective. Consequently, there is a need to develop alternative therapeutic agents that inhibits other non-VEGF angiogenic pathways. Other angiogenesis inducers, apart from VEGF, are the fibroblast growth factors basic fibroblast growth factor (b-FGF/FGF-2), making them a potential drug target for cancer, such as melanoma [137,139]. Formononetin has been found to be a novel FGF receptor 2 inhibitor as it suppressed the sprouting of FGF2-induced micro-vessel in rat aortic rings and angiogenesis, specifically targeting the FGF receptor 2-mediated AKT signaling pathway, resulting in the attenuation of tumor growth and angiogenesis [121]. As a result, formononetin could be investigated in future studies as an angiogenesis inhibitor through targeting FGFs. 

### 3.6. Metastasis-Regulatory Effects 

When highly metastatic MDA-MB-231-luc and 4TI breast cancer cells were exposed to formononetin, no significant difference in the cell viability was observed as compared to untreated cells [108]. However, treatment with formononetin (2.5–40 μmol/L) reduced the migration of MDA-MB-231 and 4TI cells in a concentration-dependent manner. Furthermore, formononetin reduced the invasion of MDA-MB-231 and 4T1 cells. Invasiveness of metastatic cancer was also attenuated in the LoVo human colon cancer cell line [109]. This phenomenon may have been due to the reduced expression levels of MMPs. MMP-2 and MMP-9 levels were reduced concentration-dependently with formononetin treatment, and formononetin elevated the expression levels of tissue inhibitor of MMP-1 and MMP-2, which are negative regulators of MMPs [108], suggesting that formononetin has the ability to influence the expression levels of proteins and genes associated with proteolytic activation. 

### 3.7. Anti-Inflammatory Effects

One of the hallmarks of cancer development is chronic inflammation. It can drive tumor progression by regulating proliferation, invasion, metastasis, angiogenesis, chemoresistance, and radio-resistance of tumor cells [1,2,38,60,61,102]. NF-κB, a pro-inflammatory transcription factor, is associated with the inflammation and suppression of apoptosis, and is a key driver of different cellular processes in multiple cancer models [21,50,60,62,64]. NF-κB can translocate into the nucleus and activate numerous genes that are involved in multiple processes crucial for multistage carcinogenesis, including proliferation, invasion, and angiogenesis, upon activation by cytokines or chemotherapeutic agents. As a result, chemotherapeutics that can potentially inhibit NF-κB have a significant role in anticancer therapy [5,12,21,38,50,58,60,62,102,129]. Formononetin demonstrated an inhibitory effect on NF-κB activation, and NF-κB is a significant transcription factor for the induction of nitric oxide synthase to decrease the production of nitric oxide in vitro [140]. The inhibition of NF-κB suggests the potential crucial and therapeutic effect that formononetin may have in protecting against inflammation, which may result in cancer development. Overall, the important oncogenic and pro-inflammatory pathways affected by formononetin have been depicted in Figure 2.

### 3.8. Combinatorial Studies with Selected Chemotherapeutics

Previous clinical studies suggest that polysaccharides obtained from *Astragalus* plants can counteract the adverse effects of chemotherapeutic drugs, including a significant reduction of myelosuppression in cancer patients [141]. In addition to the antiproliferative and proapoptotic properties of formononetin, the expression level of p53 was concentration-dependently upregulated after treatment with formononetin through increased phosphorylation of p53 at Ser15 and Ser20, enhancing its transcriptional activity [96]. For example, formononetin demonstrated synergy when coupled with the use of other chemotherapeutic drugs. Temozolomide (TMZ) is an oral chemotherapy drug often used in the treatment for certain brain cancers, such as glioblastoma multiforme. However, TMZ is a chemotherapeutic drug that is known to causes adverse side effects, including hematologic complications and both intrinsic and acquired resistance [110,142]. Although results found that both formononetin and TMZ alone were sufficient to inhibit the growth of C6 glioma cells concentration-dependently, when formononetin is used in combination with TMZ, it displayed a synergistic effect on C6 cells. The combination of both drugs increased Bax protein expression and cleaved caspase-3 and caspase-9, attenuated Bcl-2 expression, and promoted tumor cell apoptosis [110]. Furthermore, the drug combination prevented the migration of C6 glioma cells due to the down-regulated expression of MMP-2 and MMP-9. This suggests the potential use of formononetin as a combination drug during chemotherapy or as a post-operative adjuvant therapy to curb the adverse effects displayed by other chemotherapeutic drugs, such as TMZ. Co-treatment consisting of formononetin with synthetic inhibitors for ovarian cancer, such as LY294002 (PI3K inhibitor) or U0126 [mitogen activated protein kinase kinase (MEK) inhibitor], further prevented the proliferative effects on ovarian cancer cells (ES2 and OV90), thus increasing the occurrence of apoptosis in both cell lines [99]. 

Furthermore, a potent chemotherapy drug doxorubicin has been identified to induce epithelial-mesenchymal transition (EMT) in glioma cells via elevated expression of vimentin and reduced expression of E-cadherin in U87MG glioma cells [111]. Numerous studies have suggested the important role that EMT holds in carcinogenicity, metastasis, progression, and acquired chemoresistance [129,143,144,145,146]. Formononetin has been proven to be able to sensitize glioma cells to doxorubicin, and combination therapy using doxorubicin and formononetin is able to successfully reverse the induction of EMT by doxorubicin. In addition, histone deacetylase 5 (HDAC5) has been identified to enhance glioma cell proliferation, and doxorubicin-treated glioma cells have been identified to have significantly increased HDAC5 levels. However, co-treatment with formononetin reduced the expression of HDAC5 in glioma cells. These results implied that co-treatment with formononetin and other chemotherapeutic drugs, such as doxorubicin, potentially sensitizes cancer cells, such as glioma cells, through the prevention of EMT and the inhibition of HDAC5 [111]. This further suggests that formononetin could be considered as an adjuvant agent with existing chemotherapeutic drugs. 

### 3.9. Novel Semi-Synthetic Hybrids of Formononetin

Several new bioactive derivatives of formononetin have been shown to have potent anti-cancer activity. Ren et al., 2012 showed that formononetin nitrogen mustard derivative (IC50-3.8 µM) exhibited potent antitumor activity against colorectal HCT-116 cells and was associated with G2/M phase cell cycle arrest and induction of apoptosis [147]. In another study, formononetin-dithiocarbamate hybrid (IC50-1.97 µM) inhibited androgen independent prostate cancer PC3 cell growth and induced apoptosis by modulating the MAPK and wingless (Wnt) signaling pathway [148]. Several lines of evidence suggest that epidermal growth factor receptor (EGFR) is an attractive target for non-small cell lung cancer (NSCLC) therapy. Lin et al 2017 reported on a new series of formononetin derivatives following the binding model of lapatinib to EGFR. Formononetin derivatives exhibited potent anti-proliferative activity against triple negative breast cancer MDA-MB-231 cells and induced apoptosis by down-regulating multiple EGFR/PI3K/Akt/Bcl-2-associated death promoter (Bad), EGFR/ERK and EGFR/PI3K/Akt/β-catenin signaling pathways in breast cancer cells [149]. In another study a formononetin 7-phosphoramidate derivative significantly induced early apoptosis in HepG-2 cells [150]. In a recent study by Chengli et al. [151] three new derivatives of formononetin were shown to inhibit the growth, invasion and migration of A549 lung cancer cells. Furthermore, Bohong et al. [152] demonstrated that multiwalled carbon nanotube–formononetin (MWCNT-FMN) composite for sustained delivery induced apoptosis by activating reactive oxygen species (ROS) production in cervical carcinoma HeLa cells. 

## 4. In Vivo Anticancer Pharmacological Activities of Formononetin

The anticancer potential of formononetin has been demonstrated in numerous in vivo models as shown in Table 2. First, formononetin has demonstrated the potential to inhibit tumor proliferation in various murine models of cancers. As compared to controls, formononetin is able to significantly and dose-dependently inhibit local tumor growth in nude mice bearing MCF-7 human breast cancer [91]. The weight of the tumor in the formononetin-treated group was reduced significantly when compared to the control, and this was substantiated with a reduction of 39.6% in tumor weight. Similar effects have been observed using human colorectal cancer cell lines. HCT-116 nude mice xenografts (CCL-247) treated with formononetin exhibited a large reduction in tumor volume and the number of proliferating cells as compared to the vehicle-treated group [109]. Mortality and significant changes in body weight were not observed in the formononetin-treated HCT-116 nude mice xenografts, indicating that formononetin displayed tolerable toxicity while being able to significantly reduce tumor size and mass in vivo. This observation was substantiated with comparable white blood cell counts between the formononetin-treated and control group, suggesting that formononetin did not exhibit neutropenic effects in preclinical models, unlike other common chemotherapeutic drugs [109]. Furthermore, formononetin has been proven to successfully inhibit the growth of tumors in vivo through targeting the TNF-α/NF-κB pathway in colorectal tumor bearing nude mice [101], and both tumor weight and volume decreased dose-dependently after treatment with formononetin.

The antitumor potential of formononetin is also observed in human multiple myeloma xenografts in nude mice, with a decrease in Ki-67 expression levels in tumor tissues [49]. Oxidative stress induced through formononetin treatment impeded the expression of phosphorylated STAT3 and STAT5 through the STAT3 and STAT5 signaling axis by successfully removing the binding ability of both transcription factors; thereby reversing their activation and consequently suppressing tumor growth in multiple myeloma [49]. Similarly, the antiproliferative effect was also observed in vivo in U2OS osteosarcoma and PC-3 and DU-145 prostate cancer nude mouse xenografts in a dose-dependent manner through a significant reduction in tumor weight [95,105]. This further affirms the potential application of formononetin as a cancer therapeutic agent. However, it is important to note that varying concentrations of phytoestrogens may exude different effects on the proliferative and apoptotic properties of different cancer models. 

In vivo mouse metastasis models were used to study the inhibitory effects of formononetin on metastatic breast cancer cells (MDA-MB-231-luc and 4T1) for lung metastasis [108]. Formononetin is able to successfully diminish the development of lung metastasis in mouse xenografts with metastatic breast cancer, which suggests that formononetin may possess antimigration and invasion properties on breast cancer cells and therefore, can increase survival time in preclinical models. This is an important finding that could promote the use of formononetin in clinical settings for chemotherapy because it plays a protective role against breast cancer metastasis. 

Formononetin has been investigated for its ability to exhibit antioxidant properties in in vivo models. Antioxidant and estrogenic properties are important factors that could catalyze the development of cancers as they are found to be involved in oxidative damage to the cellular macromolecules. Several antioxidant enzymes, including superoxide dismutase (SOD), catalase (CAT), and glutathione peroxidase (GSH-Px), are essential to counter the reactive oxygen species in vivo [153]. Preclinical studies using ovariectomized mice supported the use of formononetin as a potent free radical scavenging molecule that can prevent lipid peroxidation, which is a crucial factor in the progression of cancer [154]. 

The pharmacological property of formononetin in vivo is dependent on the level of estrogen present, which can be proestrogenic or antiestrogenic [92]. As previously mentioned, free radicals are highly reactive chemicals that can damage several major components of cells, especially the DNA, playing a pivotal role in the development of cancer. The oxidative effects of formononetin were monitored through the levels of SOD, GSH-Px, CAT, and malondialdehyde (MDA). A high and low formononetin intake for 6 months was found to increase uterine weight and the levels of SOD, GSH-Px, and CAT, and reduced MDA content in preclinical conditions, suggesting that formononetin displayed obvious antioxidant and estrogenic effects in in vivo conditions, and the estrogenic property of formononetin is not dose-dependent [153]. 

Further studies conducted to analyze the estrogenic effects of formononetin showed that it significantly elevated the expression of atrial ER subtype ERβ in ovariectomized mice [155]. Although it has not been yet demonstrated that formononetin is able to negatively impact the ER in preclinical models, this study provides us with insight that formononetin could likely affect the regulation of ERs, which are essential in the development of hormone-sensitive cancers, such as breast and ovarian cancer.

Tumor angiogenesis is a crucial process for tumor growth and metastasis for the development of cancer. The most well-established effect is the role of VEGF and VEGFR2. However, as mentioned earlier, there is evidence that the tumor can become resistant to VEGF-targeted therapy and acquire resistance against monotherapy with VEGFR inhibitors that manipulate other angiogenesis mechanisms [156]. The antiangiogenic property of formononetin is observed in nude mouse xenografts through the reduced levels of VEGF in drug-treated animals when compared against controls [109]. In preclinical models, formononetin displayed antiangiogenic properties in a myriad of cancers, especially colorectal cancer (HCT-116) [109] and breast cancer (MDA-MB-231-luc) [108]. The HCT-116 human metastatic colorectal adenocarcinoma cell line is known for its invasiveness and metastatic properties. Following treatment with formononetin, HCT-116 tumor xenografts in mice displayed a notable reduction in the number of invaded cells [109]. In another study, formononetin at doses of 25, 50 or 100 mg/kg administered for 14 days significantly suppressed the growth of subcutaneously implanted osteogenic sarcoma U2OS tumor growth and was associated with the downregulation of miR-375 with concomitant upregulation of Bax, Caspase-3, and Apaf-1 [114]. Further studies in preclinical models with small molecules have been of interest in recent years as a therapeutic measure against cancer since multi-kinase inhibitors targeting VEGFR have been proven effective in clinical settings against breast tumors. However, these agents displayed signs of toxicity and did not exhibit high response rates. Formononetin derivative when administered intraperitoneally (i.p.) for 21 days (5 mg/kg) was shown to significantly inhibit breast tumor growth in a nude mice model when compared to EGFR inhibitors gefitinib or lapatinib and was found to be well tolerated with no significant change in the body weight of the mice [149]. 

Further research in the preclinical setting has found evidence that formononetin could be effective for antiangiogenesis treatment against human breast cancer (MDA-MB-231) xenografted in nude mice by preventing tumor growth via inhibition of tumor angiogenesis [121]. Furthermore, formononetin supplemented the effect of sunitinib, a receptor tyrosine kinase inhibitor that targets VEGFR2, on tumor growth inhibition through largely decreasing the invasiveness of cancer cells stimulated by tumor growth in vivo through the fibroblast growth factor receptor 2 (FGFR2)-mediated AKT signaling pathway, attenuating tumor growth and angiogenesis [121,157]. The inhibition of angiogenesis in xenografted human breast cancer (MDA-MB-231) by formononetin was found to modulate FGF2-induced micro-vessel growth that suppressed the emergence of rat aortic rings and angiogenesis through the repression of FGF2-initiated activation of FGFR2 and AKT signaling [121]. 

## 5. Formononetin in Clinical Studies 

Formononetin has been tested in preclinical settings for other diseases, such as Alzheimer’s [158], but clinical studies for the use of formononetin as a treatment method for cancer have yet to be conducted. Jarred and colleagues [159] identified that treatment with a red clover-derived isoflavone mixture was able to induce apoptosis in low- to moderate-grade human prostate carcinoma. Further clinical research needs to be performed to ascertain the anti-cancer properties of formononetin in multiple cancer types. 

## 6. Conclusion and Future Perspectives 

Evidence presented in this review provides a comprehensive summary on the potential anticancer properties of formononetin in both in vitro and in vivo studies and the current progress of clinical studies. Numerous molecular targets and mechanisms of actions are involved in the antitumorigenic property (primarily on the induction of cell apoptosis and the inhibition of cell proliferation) of formononetin as evidenced from numerous in vitro studies, whereas the safety and efficacy of formononetin and its metabolites in biological systems are further confirmed in in vivo studies. The tumor-inhibitory effects of formononetin have been associated with the modulation of PI3K/AKT and STAT3 signaling pathways in both in vitro and in vivo models. The various anticancer molecular targets of formononetin are briefly summarized in Figure 3. In addition, formononetin has been found to possess additive and synergistic effects with chemotherapeutic drugs, such as Sunitinib. The potential role for formononetin to be used as an adjunct therapy and future prospective drug development for cancer patients are supported through these findings. However, further studies need to be conducted (both in vivo and clinical studies) to allow for further assessment of the efficacy and safety of formononetin for prevention and treatment of various cancer types. This is crucial as various formononetin derivatives and metabolites have varying pharmacokinetic properties and activities that need to be fully elucidated, and this requires further investigation to ensure that this bioactive phytochemical is safe for clinical development. The significant antitumor properties of formononetin make it a novel candidate for anticancer drug development. 

## Figures and Tables

**Figure 1 cancers-11-00611-f001:**
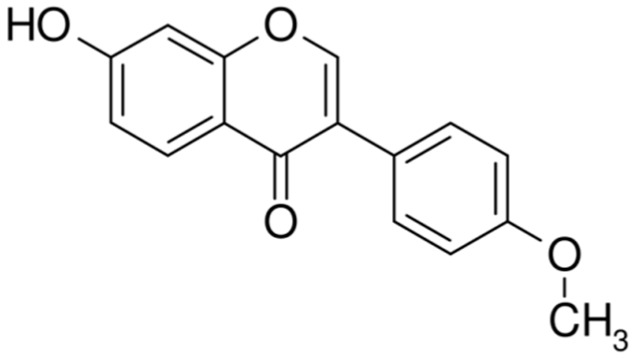
The chemical structure of formononetin.

**Figure 2 cancers-11-00611-f002:**
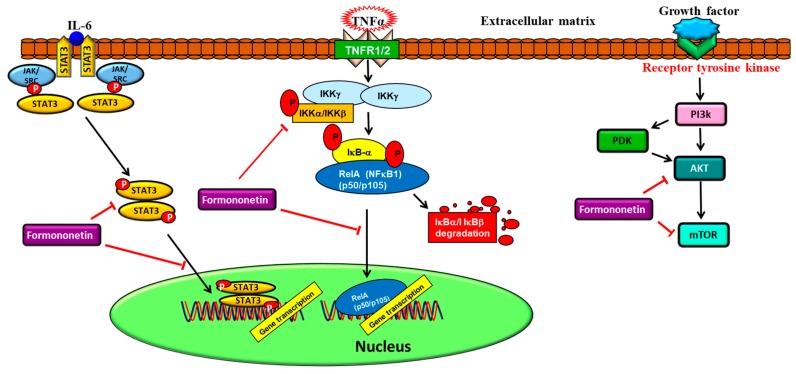
Formononetin regulates major oncogenic pathways involved in cancer progression. ⊥, inhibition/downregulation; ↑ upregulation/activation.

**Figure 3 cancers-11-00611-f003:**
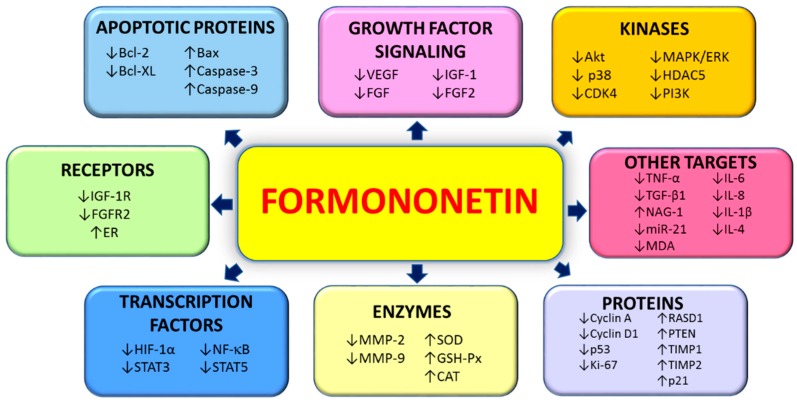
Molecular targets influenced by formononetin in cancer.

**Table 1 cancers-11-00611-t001:** In vitro anticancer effects of formononetin.

Cancer Type/Cell Line Used	Concentration	Anticancer Effect	Mechanisms of Action	References
*Bladder cancer*				
T24 cell line	50–200 μM	Antiproliferative Anti-invasion	↑Apoptosis; ↑PTEN; ↓miR-21; ↓pAKT	[98]
MCF-7 cell line	30–100 μM	Antiproliferative	↑Apoptosis; ↑G0/G1 cell cycle arrest; ↓IGF-1/IGFR-PI3K/AKT pathway	[91]
*Breast cancer*				
ER-positive MCF-7 cells and T47D cell	25–100 μM	Antiproliferative	↑Apoptosis; ↓p38MAPK pathway	[92]
ER-positive MCF-7 cells and T47D cell	25–100 μM	Antiproliferative	↑Caspase-3; ↓IGF1R; ↓miR375	[93]
MDA-MB-231 4TI	2.5–40 μmol/L	Antiproliferative	↓MMP-2; ↓MMP-9, ↓TIMP1; ↓TIMP2; ↓PI3K/AKT pathway	[108]
*Cervical cancer*				
HeLa cells	Not available	Antiproliferative	↑Apoptosis; ↓PI3K/AKT pathway; ↓ERK pathway	[97]
*Colon cancer*				
LoVo	50 μM	Anti-invasion	↑Apoptosis; ↓VEGF; ↓MMP	[109]
*Colorectal cancer*				
HCT116 cell line	6.25–200 μM	Antiproliferative	↑Apoptosis; ↑Bax; ↑NAG-1; ↓Bcl-2; ↓Bcl-xL	[37]
SW1116 cell line HCT116 cell line	20–200 μM	Antiproliferative	↑miR-149; ↓EphB3; ↓PI3K/AKT pathway; ↓STAT3 pathway	[85]
RKO cell line	20–80 μM	Antiproliferative	↑Apoptosis; ↓ERK pathway	[101]
*Glioma*				
Glioma C6 cell line	20–320 μM	Antiproliferative	↑Apoptosis; ↑Bax; ↑cleaved caspase-3 & caspase-9; ↓Bcl-2; ↓MMP-2; ↓MMP-9	[110]
*Glioblastoma*				
U87MG cell line U251MG cell line T98G cell line	50–200 μM	Antiproliferative	↓HDAC5; ↓doxorubicin-induced EMT	[111]
*Multiple myeloma*				
U266 cell line	5–60 μM	Antiproliferative	↓HIF-1α; ↓inflammatory cytokines; ↓AKT pathway	[100]
*Nasopharyngeal carcinoma*				
CNE1 cell line CNE2 cell line	5–40 μM	↓Cell viability	↑Apoptosis; ↓PI3K/AKT pathway; ↓ERK pathway	[112]
*Non-small cell lung cancer*				
A549 cell line NCI-H23 cell line	100–200 μM	Antiproliferative	↑Apoptosis; ↑caspase-3; ↑Bax; ↓Bcl-2; ↑P21; ↓cyclin A; ↓cyclin D1; ↑G1 cell cycle arrest; ↑p53 at Ser15 and Ser20	[96]
*Osteosarcoma*				
U2OS	20–80 μM	Antiproliferative	↑Apoptosis; ↑caspase-3; ↑Bax; ↓Bcl-2; ↓PI3K/AKT pathway; ↓ERK pathway	[90]
*Ovarian cancer*				
ES2 cell line OV90 cell line	20–40 µM	Antiproliferative	↑Apoptosis; ↑G0/G1 cell cycle arrest	[99]
*Prostate cancer*				
LNCaP cell line PC-3 cell line	20–80 µM	Antiproliferative	↑Apoptosis; ↑G1 cell cycle arrest; ↓AKT/cyclin D1/CDK4; ↓ERK1/2 pathway	[94]
PC-3 cell line DU-145 cell line	10–100 µM	Antiproliferative	↑Apoptosis; ↑G1/S cell cycle arrest; ↓IGF/IGFR1 pathway	[95]
PC-3 cell line	25–100 µM	Antiproliferative	↑Apoptosis; ↓IGF/IGFR1 pathway	[104]
DU-145 cell line	6.25–200 μM	Antiproliferative	↑Apoptosis; ↑Bax; ↑RASD1; ↑caspase-3; ↑PARP; ↓Bcl-2	[106]
PC-3 cell line	25–100 μM	Antiproliferative	↑Bax/Bcl-2 ratio; ↓p38MAPK/AKT pathway	[107]

AKT, protein kinase B; Bax, Bcl-2-associated protein; Bcl-2, B-cell lymphoma 2; Bcl-xL, B-cell lymphoma-extra-large; CDK, cyclin-dependent kinase; EMT, epithelial-mesenchymal transition; ERK, extracellular signal regulated kinase; EphB3, ephrin type-B receptor 3; HDAC5, histone deacetylase 5; HIF-1α, hypoxia-inducible factor 1α; IGF-1, insulin-like growth factor 1; IGF-1R, insulin-like growth factor 1 receptor; MAPK, mitogen-activated protein kinase; miR, microRNA; MMP, matrix metalloproteinase; NAG-1, NSAID-activated gene; PI3K, phosphatidylinositol 3-kinase; PTEN, phosphatase and tensin homolog; p21, cyclin-dependent kinase inhibitor; PARP, poly-ADP ribose polymerase; RASD, ras-related dexamethasone induced; STAT3, signal transducer and activator of transcription 3; TIMP, tissue inhibitor of metalloproteinase; VEGF, vascular endothelial growth factor.

**Table 2 cancers-11-00611-t002:** In vivo anticancer effects of formononetin.

Cancer Model	Dose, Duration and Route of Administration	Observed Effects	Mechanisms	References
MCF-7 cells-induced xenograft in Balb/c nude mice	60 mg/kg/day; 20 days; i.p.	↓Tumor growth	↓IGF-1/IGFR-PI3K/AKT pathway	[91]
MDA-MB-231 cells-induced xenografts in Balb/c nude mice	100 mg/kg/day; 25 days; intra-gastric (i.g.)	↓Tumor growth (synergistic effect with sunitinib)	↓FGF2-induced angiogenesis; ↓PI3K/AKT pathway; ↓STAT3 pathway	[121]
MDA-MB-231-luc cells-induced xenografts in Balb/c nude mice	10 or 20 mg/kg/day; once every 2 days for 35 days; i.p.	↓Tumor growth; ↓lung metastasis; ↑overall survival	↓PI3K/AKT pathway	[108]
HCT-116 cells-induced xenografts in Balb/c nu/nu mice	20 mg/kg/day; 2 weeks; i.p.	↓Tumor growth	↓Tumor angiogenesis; ↓VEGF	[109]
Human multiple myeloma U266 xenograft in Balb/c nude mice	20 and 50 mg/kg/day; 25 days; i.g.	↓Tumor growth	↓PI3K/AKT pathway	[92]
Human multiple myeloma tumor tissues implanted in athymic nu/nu mice	40 mg/kg; thrice/week for 3 weeks; i.p.	↓Tumor growth	↓Tumor angiogenesis; ↓STAT3/5 pathway; ↓VEGF	[49]
PC-3 cells-induced prostate xenograft in nude mice	60 mg/kg/day; 20 days; i.p.	↓Tumor growth	↑Apoptosis; ↓IGF/IGFR1 pathway	[95]
RKO tumor-bearing Balb/c nude mice	5, 10 or 20 mg/kg/day; 14 days; i.g.	↓Tumor growth	↓IL6; ↓TNF-α; ↓NF-κB pathway	[101]

AKT, protein kinase B; FGF2, fibroblast growth factor 2 receptor; IGF-1, insulin-like growth factor 1; IGF-1R, insulin-like growth factor 1 receptor; IL-6, interleukin-6; NF-κB, nuclear factor-κB; PI3K, phosphatidylinositol 3-kinase; STAT3, signal transducer and activator of transcription 3; TNF-α, tumor necrosis factor-α; VEGF, vascular endothelial growth factor.

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
