# Peer review of "Focus on Formononetin: Anticancer Potential and Molecular Targets"

_cancers, 2019, doi:10.3390/cancers11050611_

Round 1

Reviewer 1 Report

The present review by  Samantha Kah Ling Ong  et al. surely gives a very complete panorama on the possible anticancer potential of  Formononetin.

However, I think that the author’s conclusions (lines 528-30) are far too optimistic. It is mandatory, in my opinion, to clearly state (in particular in the ”Conclusion" section, but also in the "Introduction") how this compound, together with several well supported positive actions as anticancer adjuvant, presents also numerous different effects (cfr. lines 106-108, 122-127, 135-138, 140-142, 150-153,  246-260, 296-306, 314-319)  that may represent very serious difficulties for its use in cancer therapy.  It is thus not a surprise, in this context, the absence of any clinical study (line 516).

In addition, even if English is not my mother language, I feel that the manuscript should be professionally revised.  

Author Response

Query: The present review by Samantha Kah Ling Ong  et al. surely gives a very complete panorama on the possible anticancer potential of  Formononetin.

Response: We thank the reviewer for the positive comments.

Query: However, I think that the author’s conclusions (lines 528-30) are far too optimistic. It is mandatory, in my opinion, to clearly state (in particular in the ”Conclusion" section, but also in the "Introduction") how this compound, together with several well supported positive actions as anticancer adjuvant, presents also numerous different effects (cfr. lines 106-108, 122-127, 135-138, 140-142, 150-153,  246-260, 296-306, 314-319)  that may represent very serious difficulties for its use in cancer therapy.  It is thus not a surprise, in this context, the absence of any clinical study (line 516).

Response:  We have now thoroughly revised all the suggested sections as follows:

Page 5, line 120 to page 6, line 122; page 6, line 138-141; page 6, line 144 to page 7, line 149; page 7, line 171 to page 8, line 174; page 10, lines 237-248; page 12, lines 283-286; lines 291-297; page 20, lines 494-496.

Query: In addition, even if English is not my mother language, I feel that the manuscript should be professionally revised. 

Response: The manuscript has been subjected to an extensive language improvement.

Reviewer 2 Report

The authors present very interesting review concerning anticancer effect of Formononetin. The manuscript is generally well written, however moderate English changes are needed. 

Table 1 needs to be completed - e.g. duration is given only at some cell lines.

Author Response

Query: The authors present very interesting review concerning anticancer effect of Formononetin. The manuscript is generally well written, however moderate English changes are needed.

Response: We have now revised the manuscript for English language

Query: Table 1 needs to be completed - e.g. duration is given only at some cell lines.

Response: Thank you for the positive comments. We have now revised Table 1 with a uniform style.

Reviewer 3 Report

    My pleasure to review the manuscript. I have raised the following points for further improvement of the manuscript.

Line 40 and 41. This sentence should incorporate targeted therapy and immunotherapy.

Line 42: This statement is incorrect. Incidence of cancer is not linked to anticancer therapy.

Line 87, bortezomib, not 'bortozomib'.

Contradictory statements are obvious in the manuscript: for example,
Line 135 to 138, the authors state that: 'Formononetin has been found to exhibit variable degrees of ER agonism in a concentration-dependent manner (0.5-500 μM) and stimulate the proliferation the human breast cancer (MCF-7) and osteosarcoma (MG-63) cells with concomitant increase in alkaline phosphatase activity [92].' However, in Line 162 and Table 1: the authors state a contradictory statement saying: 'As compared to other isoflavones, formononetin has been proven to possess the greatest antiproliferative activity [94]. The antiproliferative property of formononetin has been observed in ER-positive breast cancer cells, such as MCF-7 and T-47D [95,96], ...' And then on Line 302: The authors state that 'This suggests that formononetin could potentially induce the proliferative property of MCF-7 breast cancer cells and exert estrogenic effects, and this may hinder its use as a potent anticancer agent, especially for hormone-sensitive cancers, such as breast cancer.' Later on, again, on Lines 419-422, the authors state that 'First, formononetin has demonstrated the potential to inhibit tumor proliferation in various murine models of cancers. As compared to controls, formononetin is able to significantly and dose-dependently inhibit local tumor growth in nude mice bearing MCF-7 human breast cancer [95].' A reader has become completely confused with these contradictory information. 

Redundancy of information scattered hear and there in the article precludes the grasp of knowledge for a reader.

Line 371: As a matter of fact, temozolomide is not a highly toxic chemotherapy drug. 

Line 384: the statement 'a potent chemotherapy drug doxorubicin (otherwise known as anthracycline),' is not correct. Doxorubicin is one of the anthracyclines.

Author Response

My pleasure to review the manuscript. I have raised the following points for further improvement of the manuscript.

Query: Line 40 and 41. This sentence should incorporate targeted therapy and immunotherapy.

Response: This sentence is now revised as suggested (page 3, line 53).

Query: Line 42: This statement is incorrect. Incidence of cancer is not linked to anticancer therapy.

Response: This statement is now revised as suggested (page 3, line 54-55).

Query: Line 87, bortezomib, not 'bortozomib'.

Response: We have made the corrections as suggested (page 5, line 101).

Query: Contradictory statements are obvious in the manuscript: for example,

Line 135 to 138, the authors state that: 'Formononetin has been found to exhibit variable degrees of ER agonism in a concentration-dependent manner (0.5-500 μM) and stimulate the proliferation the human breast cancer (MCF-7) and osteosarcoma (MG-63) cells with concomitant increase in alkaline phosphatase activity [92].' However, in Line 162 and Table 1: the authors state a contradictory statement saying: 'As compared to other isoflavones, formononetin has been proven to possess the greatest antiproliferative activity [94]. The antiproliferative property of formononetin has been observed in ER-positive breast cancer cells, such as MCF-7 and T-47D [95,96], ...' And then on Line 302: The authors state that 'This suggests that formononetin could potentially induce the proliferative property of MCF-7 breast cancer cells and exert estrogenic effects, and this may hinder its use as a potent anticancer agent, especially for hormone-sensitive cancers, such as breast cancer.' Later on,again, on Lines 419-422, the authors state that 'First, formononetin has demonstrated the potential to inhibit tumor proliferation in various murine models of cancers. As compared to controls, formononetin is able to significantly and dose-dependently inhibit local tumor growth in nude mice bearing MCF-7 human breast cancer [95].' A reader has become completely confused with these contradictory information.

Redundancy of information scattered hear and there in the article precludes the grasp of knowledge for a reader.

Response: We thank the reviewer for indicating the contradictory statements. We have now revised this section as follows:

Page 6, lines 138-141; page 12, lines 283-286; lines 291-297.

Query: Line 371: As a matter of fact, temozolomide is not a highly toxic chemotherapy drug.

Response: This sentence is now revised as suggested (page 14, lines 350-352).

Query: Line 384: the statement 'a potent chemotherapy drug doxorubicin (otherwise known as anthracycline),' is not correct. Doxorubicin is one of the anthracyclines.

Response: This sentence is now revised as suggested (page 15, lines 365-367).

Round 2

Reviewer 1 Report

The authors modified the paper according to most of the matters rised by the referee.

Reviewer 3 Report

I think this revised version of the manuscript has been improved to the level of publishable quality.